# WNK2 Inhibits Autophagic Flux in Human Glioblastoma Cell Line

**DOI:** 10.3390/cells9020485

**Published:** 2020-02-20

**Authors:** Ana Laura Vieira Alves, Angela Margarida Costa, Olga Martinho, Vinicius Duval da Silva, Peter Jordan, Viviane Aline Oliveira Silva, Rui Manuel Reis

**Affiliations:** 1Molecular Oncology Research Center, Barretos Cancer Hospital, 14784 400 Barretos, Brazil; alves.anav@gmail.com (A.L.V.A.); olgamartinho@med.uminho.pt (O.M.); vinids@gmail.com (V.D.d.S.); vivianeaos@gmail.com (V.A.O.S.); 2Life and Health Sciences Research Institute (ICVS), School of Medicine, University of Minho, 4710-057 Braga, Portugal; angela.amorimcosta@ineb.up.pt; 3ICVS/3B’s—PT—Government Associate Laboratory, 4806-909 Braga, Portugal; 4Department of Human Genetics, National Health Institute Doutor Ricardo Jorge, 1649-016 Lisbon, Portugal; peter.jordan@insa.min-saude.pt; 5BioISI—Biosystems & Integrative Sciences Institute, Faculty of Sciences, University of Lisbon, 1749-016 Lisbon, Portugal

**Keywords:** WNK2, glioblastoma cell line, autophagy, inhibition, autophagic flux

## Abstract

Autophagy is a cell-survival pathway with dual role in tumorigenesis, promoting either tumor survival or tumor death. *WNK2* gene, a member of the WNK (with no lysine (K)) subfamily, acts as a tumor suppressor gene in gliomas, regulating cell migration and invasion; however, its role in autophagy process is poorly explored. The *WNK2*-methylated human glioblastoma cell line A172 WT (wild type) was compared to transfected clones A172 EV (empty vector), and A172 WNK2 (WNK2 overexpression) for the evaluation of autophagy using an inhibitor (bafilomycin A1—baf A1) and an inducer (everolimus) of autophagic flux. Western blot and immunofluorescence approaches were used to monitor autophagic markers, LC3A/B and SQSTM1/p62. A172 WNK2 cells presented a significant decrease in LC3B and p62 protein levels, and in LC3A/B ratio when compared with control cells, after treatment with baf A1 + everolimus, suggesting that *WNK2* overexpression inhibits the autophagic flux in gliomas. The mTOR pathway was also evaluated under the same conditions, and the observed results suggest that the inhibition of autophagy mediated by WNK2 occurs through a mTOR-independent pathway. In conclusion, the evaluation of the autophagic process demonstrated that WNK2 inhibits the autophagic flux in glioblastoma cell line.

## 1. Introduction

Gliomas are the most common adult primary brain tumors [1]. Glioblastoma (GBM) is the highest-grade form of glioma (WHO grade IV), as well as one of the most aggressive types of cancer with rapid cellular growth, and highly invasive behavior, with a median overall survival time of 15-18 months [2]. The current therapeutic approach is surgery followed by concomitant radiotherapy and temozolomide-based chemotherapy [3,4,5]; however, despite significant advances in diagnosis and therapy in recent decades, the outcomes for high grade gliomas (WHO grade III-IV) remains unfavorable [6]. To change this scenario, a deeper understanding of glioma cancer biology is needed.

Autophagy is a catabolic mechanism that maintains cellular homeostasis. In this cellular process proteins or cytoplasmic organelles are sequestered by double-membrane vesicles known as autophagosomes [7]. Fusion of autophagosomes with lysosomes forms a structure in which intracellular degradation occurs, leading to a state of equilibrium of cellular metabolism, as well as apoptosis [8]. Uncontrolled autophagy is also a cell death mechanism that may occur either in the absence or concomitantly with signs of apoptosis [9]. The autophagy process has been found activated in many tumors and its inhibition can lead to both increased cell death and increased survival, depending on the tissue type, tumor grade, and therapy used [10,11]. Furthermore, it is known that prolonged and progressive autophagy stress can lead to cell death [12]. Thus, induction of autophagic cell death has been proposed as a possible mechanism of tumor suppression [13]. Since autophagy can be viewed as pro- or anti-tumor, depending on the context [14], the dissection of its role in gliomas, as well as the associated molecular mechanisms is of critical importance. This is particularly relevant because glioma cells usually respond to therapeutic agents that induce the autophagic process such as, TMZ and rapamycin in a clinical setting [15].

The serine/threonine kinase WNK2 (with no lysine protein kinase 2) acts as a tumor suppressor in GBM and is associated to carcinogenesis-related pathways [16,17,18]. WNK2 inhibits cell proliferation, invasion, and migration [19,20,21]. These effects are lost following epigenetic silencing by hypermethylation of *WNK2* promoter region [16,17]. However, the role of *WNK2* in cell death is still unclear and contradictory data concerning *WNK2* as an autophagic modulator has been reported [22,23]. Therefore, the present study aimed to explore the in vitro role of *WNK2* in autophagic process in gliomas.

## 2. Materials and Methods

### 2.1. Cell Lines and Cell Culture

The human glioblastoma cell lines A172 WT (wild-type), A172 EV (empty-vector), and A172 WNK2 (WNK2 overexpression) (Appendix A) were cultured in Dulbecco’s modified Eagle’s medium (DMEM) supplemented with 10% fetal bovine serum (FBS) (Sigma-Aldrich, St. Louis, MO, USA), 1% p/s (penicillin/streptomycin) (Life Technologies, Carlsbad, CA, USA) at 37 °C under humidified atmosphere containing 5% CO_2_. A172 EV and A172 WNK2 were generated and maintained as previously described [18]. The authentication of the cell lines was performed by a DNA short tandem repeat (STR) profile at the Diagnostic Laboratory at Barretos Cancer Hospital (São Paulo, Brazil), as previously described [24].

### 2.2. Cell Treatment

The A172 WT, A172 EV, and A172 WNK2 cell lines were plated in 6-well plates at a density of 6 × 10^5^ cells/well, and allowed to adhere overnight. After this period, the cells were starved for 3 h with DMEM 0.5% FBS. Next, for the control cells, the growth medium was replaced with DMEM 10% FBS. To evaluate the autophagic process, the cells were treated with Earle’s Balanced Salt Solution (EBSS) (Thermo Fisher Scientific, Waltham, MA, USA). To evaluate the autophagy process, 20 nM bafilomycin A1 (baf A1) (Sigma-Aldrich) was added to the EBSS. Furthermore, 10 nM everolimus (Sigma-Aldrich) was added, acting on the inhibition of mammalian target of rapamycin (mTOR), resulting in the induction of autophagy. Cells were incubated with the respective treatments for 4 and 6 h.

### 2.3. Immunofluorescence

Cells were plated in a 24-well plate at a density of 7.5 × 10^5^ cells/well, and allowed to adhere for at least 24 h. Subsequently, the cells were starved for 3 h with DMEM (0.5% FBS) before treatment, and then treated with 30 µM chloroquine (CQ) (Molecular Probes, Invitrogen, Eugene, OR, USA) for 16 h. Next, the cells were incubated with formaldehyde 3.7% in Dulbecco’s phosphate-buffered saline (DPBS 1X) (Sigma-Aldrich) and permeabilized with 0.2% Triton X-100 in DBPS 1X for 15 min at room temperature. The cells were incubated for 2 h with a primary LC3 rabbit polyclonal antibody (Molecular Probes, Invitrogen, Eugene, OR, USA) diluted in DPBS with 5% BSA (Bovine Serum Albumin), followed by the secondary antibody Alexa Fluor 488 (Life Technologies, Carlsbad, CA, USA) for 1 h at room temperature. Finally, the cells were labeled with HOECHST 33342 (1:2000) (Life Technologies) and phalloidin-rhodamine (1:200) (Molecular Probes, Invitrogen, Eugene, OR, USA). Images were acquired by the High Content In Cell Analyzer 2200 platform (GE Healthcare Life Sciences, Chicago, IL, USA) and quantification was performed in the Image-Pro software (Media Cybernetics, Rockville, MD, USA).

### 2.4. Transient Transfection

The pDest-mCherry-EGFP-LC3B and pDest-mCherry-GFP-p62 plasmids were kindly provided by Prof. Terje Johansen (Molecular Cancer Research group, Institute of Medical Biology, University of Tromsø, Tromsø, Norway) for transient transfection. Cells were plated in 6-well plates at a density of 2.5 × 10^5^ cells/well 24 h before transfection with plasmid using the Lipofectamine 3000 reagent (Invitrogen) according to the manufacturer’s recommendations. After 5 h, the transfection medium was replaced by a fresh culture medium, and the cells were incubated for another 24 h. Subsequently, the cells were starved with Hank’s Balanced Salt solution (HBSS) (Invitrogen) for 4 h prior to treatment with 200 µM baf A1 (Sigma Aldrich) for 24 h. After this period, the cells were labeled with HOECHST (1:2000) (Life Technologies). Images were acquired by the High Content In Cell Analyzer 2200 platform (GE Healthcare Life Sciences) and quantification was performed in the Image-Pro software (Media Cybernetics).

### 2.5. Western Blotting Analysis

Total protein from cell death and autophagy assays was analyzed by western blot as previously described [25]. The following antibodies were used: Anti-LC3A/B (1:1000), anti-SQSTM1/p62 (1:1000), anti-p-p70^S6K^ (Thr389) (1:1000), anti-p-4EBP1 (1:1000), anti-p-mTOR (Ser2448) (1:1000), α-tubulin (1:2000), and β-actin (1:2000), all purchased from Cell Signaling (Danvers, MA, USA). β-actin and α-tubulin were used as a loading control. HRP-conjugated goat anti-mouse and goat anti-rabbit (all from Cell Signaling) were used as secondary antibodies. Chemiluminescence using ECL (GE Healthcare Life Sciences) was detected on an Image Quant LAS4000 mini photo documentation system (GE Healthcare Life Sciences). The subsequent quantification was performed by Image J software version 1.52s (National Institutes of Health—https://imagej.nih.gov/ij/).

### 2.6. Statistical Analysis

Data from experiments were expressed as the mean ± standard deviation (SD) of three independent experiments. *p*-values were calculated by two-way ANOVA. Symbols indicate statistical comparisons (* *p* < 0.05, ** *p* < 0.01, *** *p* < 0.001). The aforementioned analysis was performed using GraphPad PRISM version 5 (GraphPad Software, San Diego, CA, USA).

## 3. Results

### WNK2 Inhibits Autophagic Flux

To investigate the potential effect of *WNK2* overexpression on autophagy, the A172-derived cell lines (A172 WT, A172 EV, and A172 WNK2) were evaluated for the main markers recommended by the guidelines for the use and interpretation of assays for autophagy monitoring (3rd edition) [26]. The cells were treated with baf A1 as an autophagy flux inhibitor, as well as with starvation (EBSS) and everolimus as autophagy inducers. The western blot analysis demonstrated that *WNK2* overexpression significantly decreased LC3B (microtubule-associated protein 1 light chain 3 beta) protein lipidation, and SQSTM1 (sequestosome 1)/p62 levels after 4 and 6 h of combinatorial treatments (Figure 1A,B) indicating a clear autophagic flux inhibition.

Since everolimus, an mTOR pathway inhibitor, was used in combination with baf A1 treatment, proteins belonging to this pathway were evaluated. Interestingly, under these conditions, the activity of mTOR or the phosphorylation of its substrates EBP1 and p70S6K did not differ between the WNK2-overexpressing cell line compared to the WT and EV controls (Figure 2A,B), suggesting that the observed autophagic flux inhibition in the WNK2 presence is mTOR-independent.

To confirm the autophagy flux inhibition observed in the *WNK2* overexpressing cells, we transfected the three A172-derived cell lines with *tandem* conjugated pDest.mCherry-GFP-p62 and pDest.mCherry-GFP-LC3B plasmids [27] and then treated for autophagy induction with HBSS and with the autophagy flux inhibitor baf A1. In this approach, yellow signal indicates the presence of LC3B or p62 in autophagosomes, whereas red signal indicates autophagolysosomes due to loss of green fluorescence in their acidic environment. We found a marked decrease in p62-positive puncta in A172 WNK2 cells in the HBSS + baf A1 treated condition (Figure 3A). On the other hand, these cells showed no change in autophagic flux as evidenced by the number of LC3B-positive puncta when compared to control cells; however, a decrease in red dots representing autophagolysosomes was evidenced between WT and WNK2 cell line (Figure 3B).

Additionally, another approach was used for evaluating autophagic flux in the edited cell lines. In these results, *WNK2* overexpression changed LC3B levels after 16 h upon another autophagy inhibitor treatment, chloroquine at 30 μM (Figure 4A,B) when compared with WT cell line.

## 4. Discussion

The role of *WNK2* in the autophagy process is still contradictory. In this study, the assessment of the autophagic process using different methodological approaches has suggested that the presence of WNK2 inhibits the autophagic flux in glioma cell lines and it is independent of the mTOR pathway.

In the autophagic process, some important markers allow to estimate autophagic activity, such as LC3 and p62. Typically, LC3A is converted to LC3B by lipidation and is present in the formation of autophagosomes [26]. p62 protein binds to ubiquitinylated proteins, that are labeled for degradation and direct them to the lysosome. Generally, p62 levels correlate inversely with autophagic activity. However, it is unclear whether p62 is degraded only by autophagy or partially by the ubiquitin-proteasome pathway [28]. Thus, p62, as well as LC3, can be transcriptionally regulated during autophagy [29]. Western blot analysis showed a significant decrease in the levels of LC3B and p62 markers after 4 and 6 h of baf A1 treatment and everolimus. In addition, immunofluorescence results demonstrated a decrease in LC3B level after CQ treatment and p62 *puncta* after HBSS + baf A1 treatment and although no statistical difference was found in the LC3B *puncta,* there was a decrease in red dots in the overexpressing WNK2 cell line when compared to the WT cell line. Previous studies point to WNK2 acting in the early stages of autophagic flux, however, with contradictory functions. Szyniarowski et al. (2011) [23] silenced *WNK2* via siRNA (small interfering RNA) in MCF-7 human breast carcinoma cells and reported p62 accumulation, thus inhibiting autophagic flux [30]. Accumulation of this protein indicates defective maturation of autophagosomes [23], thus WNK2 would act as a positive regulator of the autophagic process. On the other hand, Guo et al. (2015) silenced *WNK2* through shRNAs (short hairpin RNA) that induced a significant increase in LC3B by immunofluorescence [22] demonstrating that WNK2 could inhibit autophagy flux. It is noteworthy that according to Yoshii et al. (2017), cells that present a low number of positive puncta in the basal condition, as we reported, and that do not suffer alterations after treatment with baf A1 are suggestive of defects in the autophagy induction process [31].

It is known that one of the major biological changes in glioma is the alteration of the PI3K (phosphatidyl inositol 3 kinase)/AKT/mTOR pathway [32]. Inhibition of mTOR may be detected as dephosphorylation of its substrates p70S6K kinase and 4EBP1 (4E binding protein 1) [33] and are correlated with the autophagic process in gliomas [34,35]. To assess the impact of this pathway, we used treatment with the autophagic inducer (everolimus), a rapamycin analog that inhibits mTOR signaling in combination with an autophagic inhibitor (baf A1). The results suggest that *WNK2*-mediated autophagy inhibition occurs independently of the mTOR pathway. Previous studies relating *WNK2* and autophagy have also contradicted the effect of this gene on mTOR activity. While Szyniarowski et al. (2011) did not find any effect on mTORC1 activity in breast cancer cell lines by analyzing the phosphorylation status its p70S6K [23], another study using chronic myeloid leukemia demonstrated suppressed autophagy by activating mTOR [22]. Furthermore, it was recently reported that *mTOR* mutations have essential implication in inducing resistance to the mTOR inhibitors preserving its activity [36], as observed in our study. One hypothesis for our findings would be that *WNK2* could promote defects in the initial autophagy flux or autophagosome maturation [22,23]. During autophagosome formation PI3K activity, an upstream element in the mTOR pathway, is required and when suppressed inhibits autophagic flux. In this sense, the mechanism used by *WNK2* to inhibit autophagic flux would be similar to the 3-methyladedine autophagy inhibitor (3-MA) [36].

TMZ is considered the most effective treatment drug for GBMs and its main mechanism of action is on autophagy [37,38]. However, depending on the cellular context, it has been shown that autophagy could lead to the development of resistance to TMZ treatment rather than cell death [39,40,41]. Recently, therapeutic molecules that inhibit autophagy such as CQ and hydroxychloroquine have been used in phase I/II clinical trials concomitant with TMZ treatment and radiation and have shown increased survival in patients diagnosed with GBM [15,42]. In this study, assessment of the autophagic process has suggested that the presence of WNK2 inhibits the autophagic flux in glioblastoma cell line. It will now be interesting to study the role of *WNK2* in autophagic vesicular trafficking in response to therapy.

## Figures and Tables

**Figure 1 cells-09-00485-f001:**
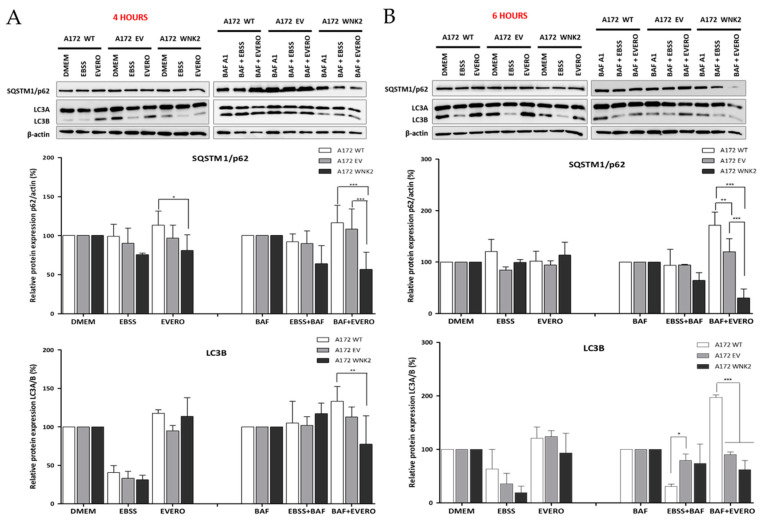
Evaluation of LC3B and p62 proteins by western blot. A172 WT, A172 EV, and A172 WNK2 cell lines were treated with bafilomycin A1 (BAF, 20 nM), starvation (EBSS medium), everolimus (EVERO, 10 nM), or the combination BAF+EVERO for 4 (**A**) or 6 h (**B**). The graphs are representative of three independent biological experiments. β-actin protein was used as an endogenous loading control. Symbols mean (*) *p* < 0.05; (**) *p* < 0.01; (***) *p* < 0.001.

**Figure 2 cells-09-00485-f002:**
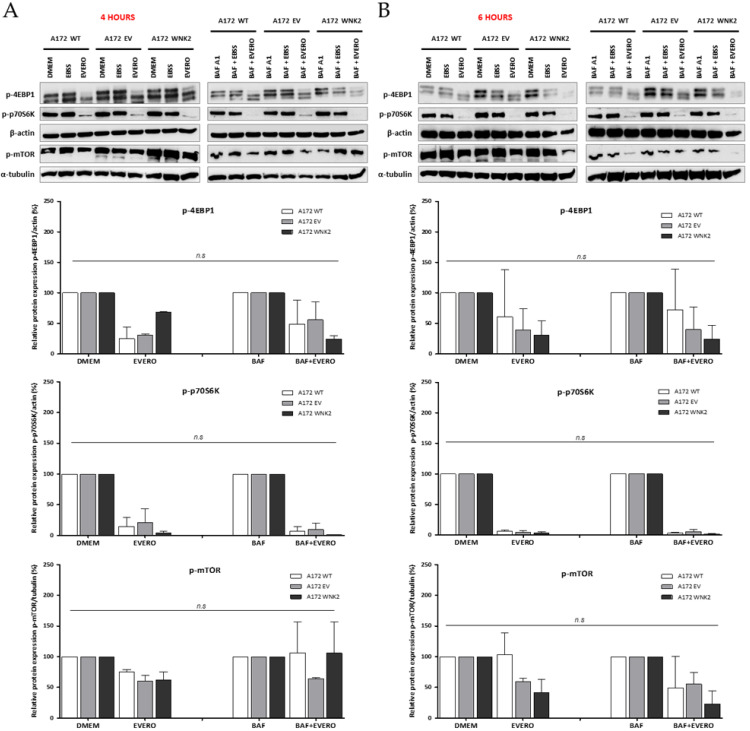
Evaluation of proteins involved in the mammalian target of rapamycin (mTOR) pathway by western blot. A172 WT, A172 EV, and A172 WNK2 cell lines were treated with bafilomycin A1 (BAF, 20 nM), starvation (EBSS medium), or everolimus (EVERO, 10 nM) for 4 (**A**) and 6 h (**B**). The protein extract was evaluated for phosphorylation of mTOR and its substrates p-p70S6K and p-4EBP1 by western blot. Normalized densitometric band intensities of mTOR activity used α-tubulin as an endogenous loading control. For the substrates p-p70S6K and p-4EBP1, β-actin was used as an endogenous control. The graphs are representative of two independent biological experiments. n.s.: Not significant.

**Figure 3 cells-09-00485-f003:**
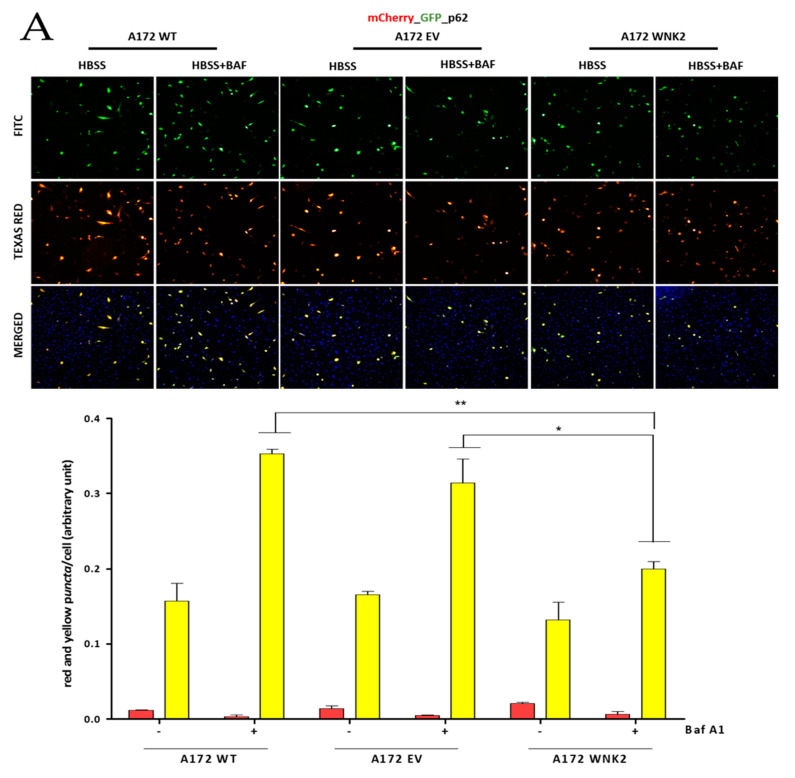
Evaluation of LC3B and p62 proteins by using transient transfection with pDest.mCherry-GFP-LC3B and pDest.mCherry-GFP-p62 plasmids. A172 WT, A172 EV, and A172 WNK2 cell lines were transfected with plasmid pDest.mCherry-GFP-LC3B (**A**) or pDest.mCherry-GFP-p62 (**B**) and then treated for 24 h with bafilomycin A1 (baf A1). Hoechst (DAPI) treatment indicates nuclear staining by blue fluorescence. FITC indicates green fluorescence and Texas Red indicates red fluorescence wavelengths by the In Cell Analyzer platform. In the figures, yellow dots indicate the presence of LC3B or p62 in autophagosomes, whereas red dots indicate autophagolysosomes due to loss of green fluorescence in an acidic environment. In the graphics, the yellow and red bars indicate the quantification of autophagossomos and autophagolysosomes, respectively, observed in the merged. The graphs are representative of two independent biological experiments. Images were quantified using Image-Pro software. Symbols mean (*) *p* < 0.05; (**) *p* < 0.01; (***) *p* < 0.001. (+) means presence of baf A1; (-) means absence of baf A1.

**Figure 4 cells-09-00485-f004:**
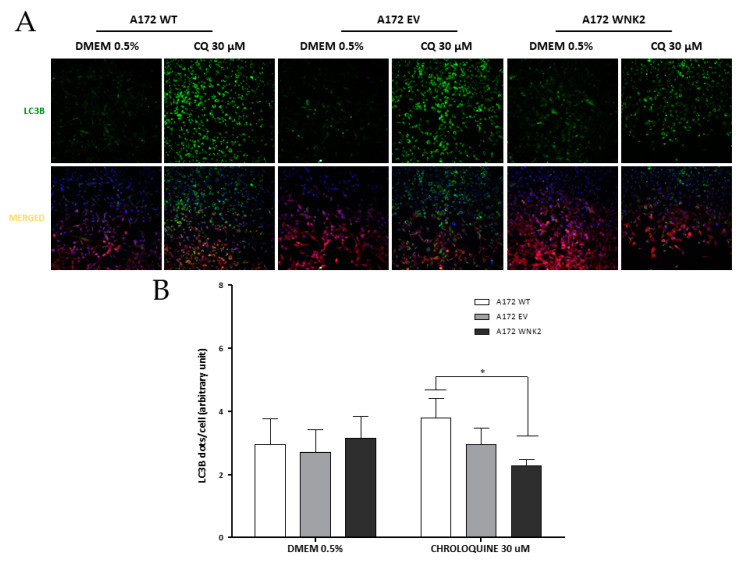
Evaluation of the effect of *WNK2* overexpression on the autophagic process after chloroquine (CQ) treatment. (**A**) A172 WT, A172 EV, and A172 WNK2 cell lines were treated with DMEM 0.5% FBS for control conditions and to induce autophagy with the autophagy flux inhibitor chloroquine (CQ, 30 μM) for 16 h after a 3-h starvation period. Images were acquired at DAPI (blue, nuclei), FITC (green, LC3 dots), and CY3 (red, cytoplasm) wavelengths by the In Cell Analyzer platform. (**B**) Graph of the quantification of LC3B vesicles in merged after control condition and treatment with CQ for 16 h. The graphs are representative of three independent biological experiments. The quantification of images was realized in Image-Pro software. Symbols mean (*) *p* < 0.05; (**) *p* < 0.01; (***) *p* < 0.001.

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
