# Peer review of "WNK2 Inhibits Autophagic Flux in Human Glioblastoma Cell Line"

_cells, 2020, doi:10.3390/cells9020485_

Round 1
Reviewer 1 Report
Authors of this manuscript reported a study of the role of WNK2 on regulating autophagy in a glioblastoma cell line. Autophagy suppression by WNK1 has been reported by Cobb’s group previously. This work thus adds to the understanding that another WNK family protein, WNK2, is associated with autophagy activation in a subset of brain tumor cells. This study used glioblastoma cell lines with epigenetically silences and overexpressed WNK2 levels to show that WNK2 overexpression reduced autophagy activation that is independent of the mTOR pathway. Although no detailed mechanism was provided, the finding may be of interest to the autophagy research field. In this manuscript, several technical issue and analyses still need to be clarified before publication is recommended. They are discussed in the following.
The WNK2 expression levels in the cell lines transfected with the empty vector and the WNK2 overexpression need to be provided. In Fig. 1A, why the starvation treatment only induces modest autophagy by decreasing the p62 in A172 WT and EV cells. The changes of the LC3A/B ratios in the starvation treatment appear in the bar plots appear to be exaggerated when compared with the western blots. Based on the western blots in Figs. 1 and 2, the empty vector transfection appears to affect the p62 levels at 4 hr and mTOR activation at 4 and 6 hrs as indicated by the darker bands. In Fig. 3, the images (FITS, Texas Red) do not appear to clearly correlate well with the cell counts in each cell. Take the example of Fig. 3A, A172 WT cells, the green and red cells are similar. Are the images presentative? Some figures showed error bars in the bar plots but other did not. The number of repeated experiments should be stated clearly.
Minor issues.
EBSS was used instead of HBSS in many places including in the figures. This needs to be corrected. Line 54, should be “relevant because glioma cells..” Line 86, should be “DPBS”. Line 115, reference should be http://rsbweb.nih.gov/ij).Author Response
Barretos, 22sd January 2020.
Dear Prof. Javier S. Castresana and Dr. Bárbara Meléndez
Re: "WNK2 inhibits autophagic flux in human glioblastoma cell line". manuscript ID (cells-694161)
Please find attached our manuscript entitled “WNK2 inhibits autophagic flux in human glioblastoma cell line". manuscript ID (cells-694161) by Alves and co-authors, that we revised for publication in Cells.
Thank you for the response to our manuscript, and for the excellent suggestions. The manuscript has been reviewed in accordance to editorial board comments/recommendation. Below, please find our detailed response.
Reviewer 1
1) The WNK2 expression levels in the cell lines transfected with the empty vector and the WNK2 overexpression need to be provided.
We agree with the reviewer`s suggestion that WNK2 expression levels of empty vector and the WNK2 overexpression must to be provided and we added a Supplementary Materials section to included it. The edited cell lines (A172 WT, A172 EV and A172 WNK2) were generated and maintained as previously described [1]. During each step of the experiments, the cell lines were validated for WNK2 gene expression. WNK2 mRNA level was evaluated by qualitative RT-PCR as showed in Figure 1. The SW1088 cell line (astrocytoma) was used as a positive expression control since this cell line does not have WNK2 promoter methylation and therefore endogenous WNK2 protein expression [1]. Regarding WNK2 protein expression, it was also evaluated by three antibodies (abcam 28852, abcam 192397 and 07-2261 Millipore), however, they did not show to be specific, giving cross-reaction with other WNK family members (data not shown).
Reference:
[1] Moniz, S.; Martinho, O.; Pinto, F.; Sousa, B.; Loureiro, C.; Oliveira, M.J.; Moita, L.F.; Honavar, M.; Pinheiro, C.; Pires, M., et al. Loss of WNK2 expression by promoter gene methylation occurs in adult gliomas and triggers Rac1-mediated tumour cell invasiveness. Human molecular genetics 2013, 22, 84-95, doi:10.1093/hmg/dds405.
2) In Fig. 1A, why the starvation treatment only induces modest autophagy by decreasing the p62 in A172 WT and EV cells.
We agree with the reviewer's comment that starvation treatment only induces modest autophagy by decreasing p62 in A172 WT and EV cells lines. The literature has been show that during autophagic flux, ubiquitin-binding adaptors, such as p62, bind to damaged organells and are sent to lysosomes for degradation [2-4], with this, studies in gliomas usually bring about the expression of the protein p62 decreased by the rapidity which the flux occurs [5,6]. Although we expected a decrease in the expression of p62 in the condition of starvation, it was not possible to identify it in our study. Unfortunately, this result does not allow us to state the mechanisms involved. It is believed that time may interfere with this result, since in some studies we observed glioma cell lines showing p62 expression in the baseline condition (DMEM), which is increased only after treatment with autophagic flux inhibitors such as bafilomycin and chloroquine [7 -9]. It is also believed that these mechanisms may be characteristic of the A172 cell line, however, it is just speculation which the signaling pathways and mechanisms involved in this process are still unclear, requiring further studies to help us elucidate this role.
Reference:
[2] Yoshii, S.R.; Mizushima, N. Monitoring and Measuring Autophagy. International journal of molecular sciences 2017, 18, doi:10.3390/ijms18091865.
[3] Sharifi, M.N.; Mowers, E.E.; Drake, L.E.; Macleod, K.F. Measuring autophagy in stressed cells. Methods in molecular biology 2015, 1292, 129-150, doi:10.1007/978-1-4939-2522-3_10.
[4] Min, Y.; Xu, W.; Liu, D.; Shen, H.; Xu, Y.; Zhang, S.; Zhang, L.; Wang, H. Earle's balanced salts solution and rapamycin differentially regulate the Bacillus Calmette-Guerin-induced maturation of human dendritic cells. Acta biochimica et biophysica Sinica 2013, 45, 162-169, doi:10.1093/abbs/gms117.
[5] Hu, S.; Wang, L.; Zhang, X.; Wu, Y.; Yang, J.; Li, J. Autophagy induces transforming growth factor-beta-dependent epithelial-mesenchymal transition in hepatocarcinoma cells through cAMP response element binding signalling. Journal of cellular and molecular medicine 2018, 22, 5518-5532, doi:10.1111/jcmm.13825.
[6] Shang, L.; Chen, S.; Du, F.; Li, S.; Zhao, L.; Wang, X. Nutrient starvation elicits an acute autophagic response mediated by Ulk1 dephosphorylation and its subsequent dissociation from AMPK. Proceedings of the National Academy of Sciences of the United States of America 2011, 108, 4788-4793, doi:10.1073/pnas.1100844108.
[7] Klionsky, D.J.; Abdelmohsen, K.; Abe, A.; Abedin, M.J.; Abeliovich, H.; Acevedo Arozena, A.; Adachi, H.; Adams, C.M.; Adams, P.D.; Adeli, K., et al. Guidelines for the use and interpretation of assays for monitoring autophagy (3rd edition). Autophagy 2016, 12, 1-222, doi:10.1080/15548627.2015.1100356.
[8] Redmann, M.; Benavides, G.A.; Berryhill, T.F.; Wani, W.Y.; Ouyang, X.; Johnson, M.S.; Ravi, S.; Barnes, S.; Darley-Usmar, V.M.; Zhang, J. Inhibition of autophagy with bafilomycin and chloroquine decreases mitochondrial quality and bioenergetic function in primary neurons. Redox biology 2017, 11, 73-81, doi:10.1016/j.redox.2016.11.004.
[9] Mauthe, M.; Orhon, I.; Rocchi, C.; Zhou, X.; Luhr, M.; Hijlkema, K.J.; Coppes, R.P.; Engedal, N.; Mari, M.; Reggiori, F. Chloroquine inhibits autophagic flux by decreasing autophagosome-lysosome fusion. Autophagy 2018, 14, 1435-1455, doi:10.1080/15548627.2018.1474314.
3) The changes of the LC3A/B ratios in the starvation treatment appear in the bar plots appear to be exaggerated when compared with the western blots.
We agree with the reviewer`s comment, however, we choose the figure with the highest representativeness of the results from three biological experiments. We believe that the error bars represent differences found between the experiments.
4) Based on the western blots in Figs. 1 and 2, the empty vector transfection appears to affect the p62 levels at 4 hr and mTOR activation at 4 and 6 hrs as indicated by the darker bands.
It is well known that the transfection process itself is capable of altering the expression of some genes since a plasmid is inserted and clones selected by antibiotic or fluorescence [10,11]. In this case, we believe that maybe the transfecting with an empty vector could have promoted some change in other ways. It is noteworthy that despite affecting p62 levels and mTOR activity when compared to WT cell line, both showed significant values in relation to WNK2 cell line, and therefore, does not invalidate our results. Regarding mTOR activity, despite the likely effect, no results were significant or indicated a possible involvement of WNK2 in the mTOR pathway and its effectors.
Reference:
[10] Vranckx, L.S.; Demeulemeester, J.; Debyser, Z.; Gijsbers, R. Towards a Safer, More Randomized Lentiviral Vector Integration Profile Exploring Artificial LEDGF Chimeras. PloS one 2016, 11, e0164167, doi:10.1371/journal.pone.0164167.
[11] Milone, M.C.; O'Doherty, U. Clinical use of lentiviral vectors. Leukemia 2018, 32, 1529-1541, doi:10.1038/s41375-018-0106-0.
5) In Fig. 3, the images (FITS, Texas Red) do not appear to clearly correlate well with the cell counts in each cell. Take the example of Fig. 3A, A172 WT cells, the green and red cells are similar. Are the images representative?
The tandem plasmids are a fluorescence-based assay in which cells are photographed at two different wavelengths (FITC, TEXAS RED). The proteins, LC3B and p62 were tandem conjugated with a mCherry protein that emits red fluorescence (TEXAS RED), and a green fluorescent protein (GFP) that emits green fluorescence (FITC). Under neutral pH conditions, which occurs in the formation of autophagosomes, both proteins (mCherry and GFP) are expressed and a yellow fluorescence (merge) can be visualized. With acidification of the medium that occurs after the fusion of the autophagosome with the lysosome-giving rise to the autolysome, the acid-sensitive GFP protein loses its fluorescence and only the red fluorescence of the mCherry protein can be observed [2,12]. Therefore, the Figure 3 shows the A172 WT, A172 EV and A172 WNK2 cell lines in starvation with HBSS and treatment with HBSS + BAF after 24 hours. One region of the well was photographed in FITC, and after in TEXAS RED, allowing the same cells to be analyzed in two fluorescence. Subsequently, the MERGE of the two fluorescence was performed to analyze yellow (mCherry + FITC) or red (mCherry) puncta. The graphs in Figures 3A and B are representative only of MERGE images from 3 experiments, in which the red bar represents red puncta identified in each treatment condition and the yellow bar represents yellow puncta. It can be observed that the number of yellow puncta increases after inhibition with bafilomycin indicating that the number of autophagosomes increases, except in cells with overexpression of WNK2. These results demonstrate that the presence of WNK2 inhibits autophagic flux in the analyzed cells.
Reference:
[2] Yoshii, S.R.; Mizushima, N. Monitoring and Measuring Autophagy. International journal of molecular sciences 2017, 18, doi:10.3390/ijms18091865.
[12] Bravo-San Pedro, J.M.; Niso-Santano, M.; Gomez-Sanchez, R.; Pizarro-Estrella, E.; Aiastui-Pujana, A.; Gorostidi, A.; Climent, V.; Lopez de Maturana, R.; Sanchez-Pernaute, R.; Lopez de Munain, A., et al. The LRRK2 G2019S mutant exacerbates basal autophagy through activation of the MEK/ERK pathway. Cellular and molecular life sciences : CMLS 2013, 70, 121-136, doi:10.1007/s00018-012-1061-y.
6) Some figures showed error bars in the bar plots but other did not.
We agree with reviewer`s comment and we apologize for the inconvenience. Unfortunately, this happen in Figure 2A and B where the edited cell lines have no error bars under conditions with EBSS and EBSS + BAF at 4 and 6 hours. In these conditions, there are only replicate of this experiment. Therefore, we change the graphs and remove these conditions. The new graph only shows DMEM, EVERO and EVERO+BAF conditions, which were obtained from three biological experiments.
7) The number of repeated experiments should be stated clearly.
We agree with the reviewer`s suggestion and changed it in accordance in the revised Figure 2, Figure 3 and Figure 4 in the article.
8) EBSS was used instead of HBSS in many places including in the figures. This needs to be corrected.
We agree with the reviewer that it seems confusing sometimes. However, in this study we used both media, EBSS and HBSS, as starvation. Both have the purpose of inducing the autophagic process [3-6]. The difference is that the HBSS saline solution needs supplementation with calcium bicarbonate to reach the ideal concentration to be used. The preparation of the solution with sodium bicarbonate is done manually which may affect the protocol and consequently in the final result analyzed. Although the literature still brings assays using HBSS, the latest update of the Guidelines for the Use and Interpretation of Assays for Monitoring Autophagy (3rd edition) recommends the use of EBSS [7]. It is a ready-to-use solution that induces autophagic flux in cells. After the beginning of some experiments our laboratory acquires this solution, and for time and methodological reasons, the plasmid assays were not repeated with the new solution keeping with HBSS. We believe that the use of both solutions has no impact on the results obtained, as the only differing immunofluorescence assay using HBSS showed results consistent with those evaluated by western blot.
Reference:
[3] Sharifi, M.N.; Mowers, E.E.; Drake, L.E.; Macleod, K.F. Measuring autophagy in stressed cells. Methods in molecular biology 2015, 1292, 129-150, doi:10.1007/978-1-4939-2522-3_10.
[4] Min, Y.; Xu, W.; Liu, D.; Shen, H.; Xu, Y.; Zhang, S.; Zhang, L.; Wang, H. Earle's balanced salts solution and rapamycin differentially regulate the Bacillus Calmette-Guerin-induced maturation of human dendritic cells. Acta biochimica et biophysica Sinica 2013, 45, 162-169, doi:10.1093/abbs/gms117.
[5] Hu, S.; Wang, L.; Zhang, X.; Wu, Y.; Yang, J.; Li, J. Autophagy induces transforming growth factor-beta-dependent epithelial-mesenchymal transition in hepatocarcinoma cells through cAMP response element binding signalling. Journal of cellular and molecular medicine 2018, 22, 5518-5532, doi:10.1111/jcmm.13825.
[6] Shang, L.; Chen, S.; Du, F.; Li, S.; Zhao, L.; Wang, X. Nutrient starvation elicits an acute autophagic response mediated by Ulk1 dephosphorylation and its subsequent dissociation from AMPK. Proceedings of the National Academy of Sciences of the United States of America 2011, 108, 4788-4793, doi:10.1073/pnas.1100844108.
[7] Klionsky, D.J.; Abdelmohsen, K.; Abe, A.; Abedin, M.J.; Abeliovich, H.; Acevedo Arozena, A.; Adachi, H.; Adams, C.M.; Adams, P.D.; Adeli, K., et al. Guidelines for the use and interpretation of assays for monitoring autophagy (3rd edition). Autophagy 2016, 12, 1-222, doi:10.1080/15548627.2015.1100356.
9) Line 54, should be “relevant because glioma cells..”
We agree with reviewer suggestion and we changed the sentences as indicated in track changes:
Line 84 has been changed to ‘’because’’ instead of ‘’once’’.
10) Line 86, should be “DPBS”.
We agree with reviewer suggestion and we changed the sentences as indicated in track changes:
Line 86 has been changed as suggested by the reviewer.
11) Line 115, reference should be http://rsbweb.nih.gov/ij).
We agree with reviewer suggestion and we changed the sentences as indicated in track changes: The reference has been changed.
Moreover, our article has been revised by professional science editor and we have formatted this revision.
We trust that this revised manuscript is now suitable for publication in Cells. Please do not hesitate to contact us if we can provide any further information.
Looking forward to hear from you,
Sincerely yours,
Rui Manuel Reis, PhD
Molecular Oncology Research Center,
Barretos Cancer Hospital,
Rua Antenor Duarte Villela, 1331
CEP 14784 400, Barretos, S. Paulo, Brazil
Phone/Fax:+551733216600 - Extension: 7090

Reviewer 2 Report
In this manuscript the authors investigate the role of WNK2 in the inhibition of autophagic flux in glioblastoma cell line.
In my opinion the authors can be review statistical analysis.
In figure 1 they should report either actin or tubulin (figure and legend)
Why didn't the authors use CQ in western blotting results?
in the different conditions examined by the authors, how does Akt vary and its active phosphorylated form?
Author Response
Barretos, 22sd January 2020.
Dear Prof. Javier S. Castresana and Dr. Bárbara Meléndez
Re: "WNK2 inhibits autophagic flux in human glioblastoma cell line". manuscript ID (cells-694161)
Please find attached our manuscript entitled “WNK2 inhibits autophagic flux in human glioblastoma cell line". manuscript ID (cells-694161) by Alves and co-authors, that we revised for publication in Cells.
Thank you for the response to our manuscript, and for the excellent suggestions. The manuscript has been reviewed in accordance to editorial board comments/recommendation. Below, please find our detailed response.
Reviewer 2
1) In my opinion the authors can be review statistical analysis.
We agree with reviewer`s comment and we reviewed. We agree that some analyzes may have caused confusion regarding the correspondence of the images and graphics as well as the error bars. This happened because in the most experiments the graphs are representatives from three biological experiments. In order to be stated clearly it we indicated the number of repeated experiments in the revised Figure 2, Figure 3 and Figure 4 as showed in track changes.
Moreover, in Figure 2A and B which the edited cell lines have no error bars under conditions with EBSS and EBSS + BAF at 4 and 6 hours. In these conditions, there is only replicate of this experiment. Therefore, we change the graphs and remove these conditions. The new graph only shows DMEM, EVERO and EVERO+BAF conditions, which were obtained from three biological experiments as indicated in track changes.
2) In figure 1 they should report either actin or tubulin (figure and legend)
We agree with reviewer`s comment and we changed as indicated in track change. In the normalization of Figure 1, b-actin was used. It has been changed and described in the caption of the figure.
3) Why didn't the authors use CQ in western blotting results?
Bafilomycin is a vacuolar H + ATPase inhibitor that controls lysosome pH (V-ATPase) by preventing autophagic flux from being completed. Similar to this, chloroquine when in contact with lysosomes is capable of altering its pH so that autophagic degradation by lysosomes is inhibited [1]. Both may be used in assays to evaluate the autophagic process, acting on a common mechanism, flux inhibition [2]. CQ was not used in western blot assays because the kit containing the drug and antibody to be used by immunofluorescence was purchased after western blot complete and was used to complement the results mentioned above, demonstrating from a new approach with similar treatment the impact of WNK2 on autophagic flux.
Reference:
[1] Redmann, M.; Benavides, G.A.; Berryhill, T.F.; Wani, W.Y.; Ouyang, X.; Johnson, M.S.; Ravi, S.; Barnes, S.; Darley-Usmar, V.M.; Zhang, J. Inhibition of autophagy with bafilomycin and chloroquine decreases mitochondrial quality and bioenergetic function in primary neurons. Redox biology 2017, 11, 73-81, doi:10.1016/j.redox.2016.11.004.
[2] Mauthe, M.; Orhon, I.; Rocchi, C.; Zhou, X.; Luhr, M.; Hijlkema, K.J.; Coppes, R.P.; Engedal, N.; Mari, M.; Reggiori, F. Chloroquine inhibits autophagic flux by decreasing autophagosome-lysosome fusion. Autophagy 2018, 14, 1435-1455, doi:10.1080/15548627.2018.1474314.
4) In the different conditions examined by the authors, how does Akt vary and its active phosphorylated form?
We agree with reviewer`s comment that would be interesting evaluated AKT status in these conditions since the mTOR pathway involves AKT upstream. However, we did not address whether the presence of WNK2 interfered with the phosphorylated form and therefore AKT activity, we only evaluated the mTOR activity and its effectors.
Moreover, our article has been revised by professional science editor and we have formatted this revision.
We trust that this revised manuscript is now suitable for publication in Cells. Please do not hesitate to contact us if we can provide any further information.
Looking forward to hear from you,
Sincerely yours,
Rui Manuel Reis, PhD
Molecular Oncology Research Center,
Barretos Cancer Hospital,
Rua Antenor Duarte Villela, 1331
CEP 14784 400, Barretos, S. Paulo, Brazil
Phone/Fax:+551733216600 - Extension: 7090

Round 2
Reviewer 1 Report
The authors have addressed all issues appropriately.
Reviewer 2 Report
After revision the manuscript has improved